# Recent Advances in Synthetic Drugs and Natural Actives Interacting with OAT3

**DOI:** 10.3390/molecules28124740

**Published:** 2023-06-13

**Authors:** Ying Chen, Hongyan Li, Ke Wang, Yousheng Wang

**Affiliations:** 1Key Laboratory of Geriatric Nutrition and Health, Ministry of Education, Beijing Technology and Business University, Beijing 100048, China; chenying@btbu.edu.cn (Y.C.); lhy@st.btbu.edu.cn (H.L.); wangke9509@163.com (K.W.); 2Rizhao Huawei Institute of Comprehensive Health Industries, Shandong Keepfit Biotech. Co., Ltd., Rizhao 276800, China

**Keywords:** organic anion transporter 3, drug–drug interactions, herb–drug interactions, natural active compounds, inhibitors

## Abstract

Organic anion transporter 3 (OAT3) is predominantly expressed in the kidney and plays a vital role in drug clearance. Consequently, co-ingestion of two OAT3 substrates may alter the pharmacokinetics of the substrate. This review summarizes drug–drug interactions (DDIs) and herbal–drug interactions (HDIs) mediated by OAT3, and inhibitors of OAT3 in natural active compounds in the past decade. This provides a valuable reference for the combined use of substrate drugs/herbs for OAT3 in clinical practice in the future and for the screening of OAT3 inhibitors to avoid harmful interactions.

## 1. Introduction

The two primary superfamilies, ATP binding cassette (ABC) and solute carrier (SLC), play a crucial role in the absorption, distribution, and elimination of various endogenous substances such as hormones and signal molecules, as well as exogenous substances and drugs [1,2]. Generally, SLC transporters mediate the influx of substances from the blood into the epithelium, while ABC transporters mediate the efflux of substances [3,4,5]. The SLC22 family comprises six primary subfamilies: organic anion transporter (OAT), OAT-like, OAT-related, organic cation transporter (OCT), organic cation and carnitine transporter (OCTN), and OCT/OCTN-related [6,7].

OAT3 is considered the primary transporter in OAT due to its broad substrate specificity. OAT3 (encoded by SLC22A8) is widely distributed in the kidney, liver, choroid plexus [8], olfactory mucosa, brain [9], retina, and placenta [10], but it is pre-dominantly expressed on the basolateral membrane of renal tubular cells [11,12]. OAT3 utilizes dicarboxylate (e.g., α-Ketoglutarate) to exchange organic anions, with the concentration gradient as the driving force [13]. It first transports organic anions (OAs) from the blood to the proximal tubules through the basolateral membrane and then discharges OAs into the urine through the apical membrane of tubular epithelial cells, reabsorbing specific compounds in the glomerular filtrate back into the internal circulation [14,15].

OAT3 plays a crucial role in the uptake, distribution, and excretion of various endogenous/exogenous substances, but current research has primarily focused on the interactions between OAT3 and clinical drugs [16,17,18]. When two or more drugs are taken simultaneously, OAT3 becomes a target of drug competition [19,20], potentially altering the toxicity [21], pharmacokinetics [22], and function of these drugs and resulting in potential drug–drug interactions (DDIs) [23,24]. However, when herbal medicines and OAT3 substrate drugs are taken together, they may lead to severe herbal drug interactions (HDIs), such as liver injury [25], increased clotting time, headache, insomnia, and even coma and death [26,27]. Research has shown that not all of the interactions mentioned above are negative. For example, enalaprilat and quinaprilat are two types of antihypertensive drugs. When used in combination with some synthetic drugs or flavonoids, their transport through OAT3 is inhibited, leading to synergistic blood pressure reduction [28,29]. OAT3 has a significant role in DDIs and HDIs [13], with effects that can be either beneficial or adverse. In this review, we summarize the DDIs and HDIs mediated by OAT3, as well as OAT3 inhibitors in natural products over the past decade. Through this review, we can predict and avoid DDIs/HDIs mediated by OAT3.

## 2. OAT3 and Synthetic Drug–Drug Interactions

The kidney is primarily responsible for drug clearance after administration [22]. Kidney drug transporters play a crucial role in drug transport between the blood and lumen [30], including antibiotics, diuretics, proton pump inhibitors, non-steroidal anti-inflammatory drugs, antiviral drugs, and anticancer drugs [31,32]. Simultaneous administration of multiple drugs can result in changes in drug levels compared to a single administration [33]. DDIs can cause higher or lower levels of drugs, thereby altering their toxicity or efficacy (Table 1) [5,34]. Table 1 summarizes the DDIs mediated by OAT3 reported in the last decade.

Imipenem is an antibiotic with antibacterial effects, but when administered alone, it is rapidly degraded by dehydropeptidase-1 or renal dipeptidase (also termed DPEP1) [44]. Cilastatin and imipenem are both substrates of OAT3. When administered together intravenously, the plasma concentration–time curve (AUC) for imipenem is significantly increased by 1.65-fold, while the plasma/serum concentration half-time (t_1/2β_) of imipenem increased by 2.35-fold. Cilastatin also inhibits the uptake of imipenem mediated by OAT3 in the kidney, thereby reducing the hydrolysis of and acting as a complement to imipenem [35,45,46].

In addition to its role in DDIs, OAT3 inhibition has been shown to have a renoprotective effect against nephrotoxic drugs [47]. For example, diclofenac [48], a commonly used non-steroidal anti-inflammatory drug (NSAID), can cause kidney damage [49]. However, when diclofenac is co-administered with cilastatin, an OAT3 inhibitor, the AUC_0–12h_ of diclofenac increases by 35.4% and its plasma clearance (CL_p_) significantly decreases. Co-administration of cilastatin with diclofenac acyl glucuronide, a metabolite of diclofenac, leads to an increase in the concentration of diclofenac acyl glucuronide in the plasma by 46.7%, but a slight decrease in AUC_0–12h_ of diclofenac acyl glucuronide in the kidney by 18.0%. Moreover, diclofenac acyl glucuronide but not diclofenac exhibited OAT-dependent cytotoxicity and was identified as an OAT1/3 substrate. The accumulation of diclofenac acyl glucuronide in primary proximal tubule cells (RPTCs) is reduced, and the cytotoxicity caused by diclofenac acyl glucuronide is also reduced [36].

Methotrexate (MTX) is a drug used for the high-dose treatment of various malignant tumors or the low-dose treatment of rheumatoid arthritis and psoriasis [50,51]. Delayed elimination of MTX can lead to severe toxicity, such as bone marrow suppression, mucositis, and kidney damage [37]. Kidney clearance is the main pathway for MTX clearance (65–80%). Proton pump inhibitors (PPIs) inhibit the hOAT3-mediated uptake of MTX with IC_50_ values of 0.40–5.5 μM [37]. Rhein, the main metabolite of diacerein, markedly inhibited MTX accumulation in rat kidney slices and hOAT3-HEK293 cells, indicating that OAT3 is involved in DDIs in the kidney. When the two drugs were co-administered orally, the maximal plasma/serum concentration (C_max_) and AUC of MTX were increased by 2.5- and 4.4-fold, respectively, and the CL of MTX was decreased by 66.7%. When the two drugs were co-administered intravenously, the t_1/2β_ of MTX was prolonged by 86.7%, the CLP was decreased by 57.6%, and the AUC was increased by 183.4%. Rhein alleviated MTX-induced renal toxicity in vivo mainly by inhibiting OAT3 to reduce the renal clearance of MTX [38]. Masahiro Iwaki et al. found that seven NSAIDs-Glu exhibit a concentration-dependent inhibitory effect on MTX uptake through OAT3, with diclofenac-Glu having the most effective inhibitory ability (IC_50_ = 3.17 μM) [39]. In addition to the above-mentioned drugs, tranilast, an anti-allergic drug, can inhibit MTX transport. 

OAT3 renal sections and HEK293T-OAT3 cell uptake experiments have shown that tranilast can inhibit MTX uptake by OAT3. Pharmacokinetic studies in rats have shown that when MTX (5 mg/kg) is administered orally alongside tranilast (10 mg/kg), the C_max_ and AUC_0–24h_ of MTX increased to 2.14- and 1.46-fold, respectively, while CL_z/f_ and V_z/f_ decreased by 24.90% and 39.23%, respectively [40].

Quinapril is an angiotensin-converting enzyme (ACE) inhibitor used to treat hypertension and congestive heart failure. In a pharmacokinetic study on rats, co-administration of quinapril (3 mg/kg) with gemcabene (30 mg/kg) resulted in a 40% decrease in the urinary excretion of quinaprilat (quinaprilat, the metabolite of the quinapril, is also the substrate of OAT3) and a 53% increase in the AUC_0–24h_ of quinaprilat, leading to a reduction in blood pressure. Subsequent studies discovered that gemcabene inhibited quinaprilat uptake by hOAT3 and rOat3 at IC_50_ values of 35 and 48 μM. Moreover, gemcabene acylglucuronide, the major metabolite of gemcabene glucuronidation, also inhibited the hOAT3- and rOat3-mediated uptake of quinaprilat at IC_50_ values of 197 and 133 μM, respectively. This indicated the mechanism by which concomitant intake of gemcabene with quinapril can reduce blood pressure [28]. In another study, Ni et al. investigated the effects of several common drugs on the OAT3-mediated uptake of enalaprilat (another oral ACE). Benzbromarone was found to be the most effective inhibitor of OAT3 (IC_50_ = 0.14 μM), while diclofenac sodium was the weakest inhibitor (IC_50_ = 6.13 μM) [29].

One study showed that concurrent treatment with mizoribine and bezafibrate can result in rhabdomyolysis. It was suggested that both bezafibrate and mizoribine are substrates of OAT3, and when bezafibrate was co-administered orally with mizoribine, the t_1/2β_ of bezafibrate increased to 1.39-fold, and the renal clearance (CL_r_) decreased to 0.81-fold. However, when bezafibrate was co-administrated with mizoribine intravenously, the AUC and t_1/2β_ of bezafibrate increased to 1.29- and 1.25-fold, respectively. This study also suggested that mizoribine can competitively inhibit the uptake of bezafibrate by OAT3 [41].

Similarly, in a study in which benzylpenicillin and acyclovir were co-administered intravenously in rats, the cumulative urinary excretion of acyclovir decreased by 45% and CL_R_ decreased by 44%, while the t_1/2β_ of acyclovir increased by 1.9-fold. This indicates that benzylpenicillin can inhibit the renal excretion of acyclovir by inhibiting OAT3 [42].

Wen et al. found that when piperacillin and tazobactam were administered simultaneously, both CL_p_ and CL_R_ of tazobactam were decreased, and the AUC, t_1/2β_, and k_m_ of tazobactam were increased to 2.15-, 1.24-, and 1.56-fold, respectively. Moreover, piperacillin can competitively inhibit the uptake of tazobactam mediated by OAT3 [43].

## 3. OAT3 and Herb–Drug Interactions (HDIs) 

Natural medicines containing flavonoids and other functional components, particularly polyphenols, are increasingly used in clinical therapy [52]. In the elderly population with chronic diseases such as hypertension, hyperlipidemia [53], and hyperuricemia, it is common to use prescription drugs, over-the-counter drugs, and natural products simultaneously [47,54]. A survey showed that 78% of elderly respondents take both prescription drugs and dietary supplements, and 32.6% of them experience potential adverse drug reactions when they were taking a combination of herbs and other drugs [55]. Some herbs may directly cause organ toxicity, alter the pharmacokinetics or efficacy of prescription drugs mediated by OAT3, or inhibit enzymes involved in drug metabolism [26].

Purarin (PUR) is an isoflavone component extracted from *Pueraria lobata* [56]. When PUR and MTX are orally administered in combination, the C_max_, AUC, and t_1/2β_ of MTX are increased by 79%, 74%, and 70%, respectively. When PUR and MTX are co-administered intravenously, the AUC and t_1/2β_ of MTX increase by 59% and 37%, respectively, and a 37% decrease in CL_p_ was observed. PUR also simultaneously inhibits MTX uptake in rat kidney slices and HEK293-OAT3 cells. Therefore, the combined administration of PUR can inhibit OAT3 and enhance MTX exposure (Table 2) [57].

Steviol glucuronide is the major metabolite of steviol, and its uptake is primarily mediated by OAT3 (Wang et al.). Inhibition studies have shown that three drugs (with IC_50_ values between 2.92 and 9.97 μM) can inhibit steviol glucuronide uptake mediated by OAT3 in a concentration-dependent manner, and may also alter its renal clearance [58]. Steviol acyl glucuronide, the main circulating metabolite of steviol glycosides after oral administration, is also a substrate of OAT3 [63]. Therefore, if the activity of OAT3 is inhibited, it may alter the pharmacokinetic characteristics of steviol acyl glucuronide. Zhou et al. showed that probenecid and glimepiride inhibit hOAT3/rOat3-mediated steviol acyl glucuronide transport in a concentration-dependent manner (IC_50_ < 5 μM) (Table 2). When administered in combination with probenecid or glimepiride, the C_max_ of steviol acyl glucuronide was increased to 10.8- or 9.7-fold, respectively. The AUC_6–8h_ of steviol acyl glucuronide was increased to 2.9- or 2.5-fold, respectively. Therefore, people with impaired renal function should be cautious when taking steviol glycoside products [59].

The increased renal transport of imipenem mediated by OAT3 can lead to certain renal toxicity. Apigenin, a flavonoid compound widely distributed in traditional Chinese medicine, is a potent OAT3 inhibitor [64]. In cases with the combined administration of apigenin and imipenem, the survival rate of hOAT3-HEK93 cells significantly increased, and the IC_50_ value of imipenem within the cells increased to 8.1-fold. At the same time, the AUC_0–6h_ of imipenem increased by 46%, while the CL_p_ and CL_r_ of imipenem decreased by 29% and 52%, respectively. The intracellular accumulation of imipenem decreased, thereby partially reducing the cytotoxicity of imipenem. These results indicate that the inhibition of apigenin on hOAT3 can protect the cytotoxicity induced by imipenem in vitro. Therefore, apigenin can be used as a potential clinical drug to alleviate adverse renal reactions caused by imipenem [60].

Natural drugs containing flavonoids are often taken concomitantly with prescribed or over-the-counter drugs, which may lead to potential interactions [65]. For instance, Ni et al. reported the inhibitory effects of various flavonoids with distinct structures on the activity of OAT3 transport drugs [66]. Among these flavonoids, galangin exhibited the strongest inhibitory effect on the activity of OAT3 with an IC_50_ value of 0.03 μM, while myricetin exhibited the weakest inhibitory effect, with an IC_50_ value of 22.58 μM (Table 2) [29].

Lu et al. [61] reported that dichloromethane extract from *Juncus effusus* can alter the pharmacokinetics of furosemide (FS), an OAT3 substrate drug, in rats. The AUC_0–t_ of FS was increased by 80% and 55% after oral or intravenous administration of *Juncus effusus* (D), respectively. Therefore, caution is needed to avoid potential side effects resulting from the interaction between OAT3 substrate drugs and *Juncus effusus* (D). Ma et al. observed a 32% and 52% increase in the AUC_0–t_ of FS following single or multiple dose co-administration of FS and rhubarb extract, respectively [62].

## 4. Natural Bioactive Inhibitors of OAT3

Many natural bioactive compounds, including flavonoids and their metabolites such as sulfates and glucuronides, have been identified as potent inhibitors and/or substrates of OAT3 [67,68]. The study of OAT3 inhibitors can help us predict and avoid potential OAT3-mediated DDIs/HDIs [69], and these inhibitors may also have potentially beneficial effects on renal diseases, such as AAI-induced kidney disease [70]. Table 3 summarizes the natural bioactive compounds identified as OAT3 inhibitors in the last decade. These compounds are classified into nine groups: phenolic acids, flavonoids, alkaloids, anthraquinones, phenols, phenylpropanoids, terpenoids, phenanthrenoids, and others.

### 4.1. Phenolic Acids

*Salvia miltiorrhiza* has been used to treat cardiovascular diseases, but its biochemical mechanisms are still unclear. However, studies by Wang et al. have shown that six hydrophilic components from *Salvia miltiorrhiza*, tanshinol, rosmarinic, lithospermic acid, salvianolic acid A, salvianolic acid B, and protocatechuic acid (Figure 1), have significant inhibitory effects on substrate uptake mediated by mOAT3. Among these, lithospermic acid (K_i_ = 31.3 μM), rosmarinic (K_i_ = 4.3 μM), and salvianolic acid A (K_i_ = 21.3 μM) produce virtually complete inhibition [79].

Previous studies have demonstrated that caffeic acid from coffee, fruits, and vegetables significantly inhibits OAT3 (IC_50_ = 5.4 μM) [71]. Yuichi et al. uncovered that caffeic acid inhibits substrate uptake by OAT3 in a concentration-dependent manner (IC_50_ < 10 μM) (Table 3) [80].

Wang et al. investigated the effects of nine compounds extracted from natural diets and herbs on hOAT3-mediated uptake. The results indicated that four of them significantly inhibited the transport of hOAT3, with 1,3-dicaffeoylquinic acid and ginkgolic acid (17:1) exhibiting 41% inhibition, while a 30–35% reduction in hOAT3 mediated uptake was observed in response to 1,5-dicaffeoylquinic acid and ginkgolic acid (15:1) [72].

### 4.2. Flavonoids

In Chinese herbal medicine, *Scutellaria baicalensis* is used to treat inflammation and hypertension. Research by Xu found that three main bioactive substances in *Scutellaria baicalensis* exhibit OAT3 inhibition. Baicalin can effectively inhibit the influx of OAT3 substrate (IC_50_ = 13.0 μM), and baicalein (IC_50_ = 2.4 μM) and wogonin (IC_50_ = 1.3 μM) can also inhibit OAT3 significantly (Figure 2) [70].

Li et al. extracted 270 substances from Chinese herbal medicine to screen OAT3 inhibitors and found 10 flavonoids that can significantly inhibit OAT3 (IC_50_ between 1.51 and 14.77 μM). Among them, eugenin, calycosin, wogonin, luteolin, quercetol, and scullcapflavone II had noncompetitive inhibitory effects on OAT3. However, oroxylin A, viscidulin III, and 3,5,7,4′-Tetra-*O*-methyl-kaempferol competitively inhibit OAT3 [73].

Another study demonstrated that epicatechin may significantly inhibit substrate transport by hOAT3 (IC_50_ > 50 μM) [72].

### 4.3. Alkaloids

Zheng et al. demonstrated that camptothecin, 10-hydroxycamptothecin, 10-methoxycamptothecin (MCPT), and 9-nitrocamptothecin (Figure 3) could significantly inhibit OAT3-mediated substrate uptake (IC_50_ < 10 μM) [74].

Tryptanthrin, an alkaloid isolated from the medicinal indigo plant *Strobilanthes cusia*, was shown to be a potent OAT3 noncompetitive inhibitor (IC_50_ = 0.93 ± 0.22 μM, Ki = 0.43 μM) [81].

### 4.4. Anthraquinones

Ma et al. evaluated five anthraquinones isolated from rhubarb (Figure 4); among them, rhein, emodin, and aloeemodin significantly inhibited the uptake of 6-carboxyl fluorescein (6-CF, substrate of OAT3) via hOAT3, while chrysophanol and physcion showed slight inhibition [62].

Wang et al. evaluated 22 substances isolated from *Semen cassiae*, and 6 anthraquinones significantly (IC_50_ < 10 μM) inhibited OAT3-mediated transport. In vivo experimental results demonstrated that *Semen cassiae* extract can nearly eliminate the alterations in renal histology induced by mercury chloride in rats. With docking and validation of OAT3 inhibitors, it may become a drug for treating Hg-induced renal injury [75].

### 4.5. Phenols

As shown in Figure 5, two phenolic compounds, 9-Dehydroxyeurotinone and 2-O-Methyl-9-dehydroxyeurotinone from extracts of *Semen cassiae* are potent inhibitors of OAT3 (IC_50_ < 10 μM). In addition, 9-dihydroxyurotinone is a noncompetitive inhibitor of OAT3 [75].

Tatsuya et al. simultaneously injected 6-CF (1 mg/kg) and EGCG (60 mg/kg) intravenously in rats and detected the concentration of 6-CF in the blood and urine. Their results demonstrated that simultaneous injection elevated the plasma concentration of 6-CF by 8-fold after 1 h, while the AUC_0–1h_ was significantly increased to 2.2-fold. In contrast, EGCG significantly reduced the CLr of 6-CF, indicating that EGCG can inhibit OAT3 [76].

Li et al. demonstrated that 2,4,6-Trichloro-3-methoxy-5-methylphenol extract from *Lilium maximowiczii* significantly competitively inhibited OAT3 (IC_50_ = 3.93 μM) [35].

### 4.6. Phenylpropanoids

Dioscorealide B is a phenylpropanoid compound extracted from *Dioscorea esculenta* that has been shown to be an effective noncompetitive inhibitor of OAT3 (IC_50_ < 10 μM). Another phenylpropanoid compound, wedelolactone, extracted from *Eclipta prostrata* L., is also an effective inhibitor of OAT3 (IC_50_ < 10 μM). Moreover, wedelolactone significantly increased serum AAI concentrations and ameliorated renal injuries in AAI-treated mice (Figure 6) [35].

### 4.7. Terpenes

There is currently only one terpene compound, 3-Oxo-16α- hydroxy-olean-12-en-28β-oic acid, which was isolated from *Eclipta prostrata* L. It was identified as an OAT3 inhibitor with an IC_50_ value of 8.05 μM, inhibiting OAT3 in a competitive manner (Figure 7) [35].

Li et al. demonstrated that ursolic acid, the major bioactive component in pomegranate, significantly inhibits transport mediated by OAT3 (IC_50_ = 18.9 μM) [77].

### 4.8. Phenanthrenoids

Li et al. identified 16 compounds from *Juncus effusus*. Among them, 7-carboxy-2-hydroxy-1-methyl-6-vinyl-9,10-dihydrophenanthrene, 7-carboxy-2-hydroxy-1-methyl-8-vinyl-9,10-dihydrophenanthrene, 6-carboxy-2-hydroxy-1-methyl-8-vinyl-9,10-dihydrophenanthrene, 7-carboxy-2-hydroxy-1-methyl-5-vinyl-9,10-dihy-drophenanthrene, and 4,4′-dihydroxy-3,3′-dimethoxybenzophenone were potent inhibitors of OAT3 (IC_50_ < 5 μM). Their structural formulas are shown in Figure 8 [78].

### 4.9. Others

Lu et al. [61] evaluated the effects on OAT3 of 172 extracts and showed that 14 were strong inhibitors of OAT3 (IC_50_ between 0.3 and 4.8 µg/mL). Generally speaking, the n-Butanol extract from plants is more effective. For example, the n-Butanol extract IC_50_ value for *Anchusa azurea*, *Symphytum asperum,* and *Echium russicum* are 0.343, 0.406, 0.460, respectively.

With regards to the effects of natural bioactive compounds, most studies used the IC_50_ value as an indicator for OAT3 inhibition. However, depending on the potential differences in protein binding and tissue-specific concentrations, IC_50_ alone may not be sufficient to indicate that exposure to the active compound can cause effective effects. Therefore, further in vivo studies are needed to verify the efficacy of these natural bioactive compounds.

In conclusion, when these substances are administered alongside other drugs or herbs, they may change the pharmacokinetics of the drug or herbal elimination and have certain clinical consequences, including reduced therapeutic efficacy or specific side effects caused by interactions. Therefore, caution should be exercised when administering multiple drugs involving the above substances.

## 5. Conclusions

OAT3-mediated DDIs/HDIs have both potential benefits and risks. On the one hand, they can enhance drug efficacy by prolonging its half-life, but on the other hand, they may lead to nephrotoxicity by increasing drug accumulation in the kidneys. Therefore, it is important to pay attention to the potential DDIs/HDIs mediated by OAT3 during drug/herb development and in clinical practice. In this regard, the identification of natural active OAT3 inhibitors, which are summarized into nine categories, can help predict and avoid potential interactions. A better understanding of the regulatory pathways involved can also lead to the development of new methods to alleviate or avoid DDIs/HDIs mediated by OAT3. While taking advantage of OAT3-mediated interactions can be beneficial in clinical treatment, it is also important to remain vigilant with respect to potential interactions with unreported drugs. The key role played by OAT3 in DDIs/HDIs has been confirmed, and this related review may also provide a reference for future clinical application of drugs as OAT3 substrates to avoid adverse effects caused by DDIs/HDIs. In addition, there is limited research on natural OAT3 inhibitors, with most studies limited to the in vitro level. The effects of and mechanisms of action for specific natural compounds on OTA3 inhibition in vivo require further research.

## Figures and Tables

**Figure 1 molecules-28-04740-f001:**
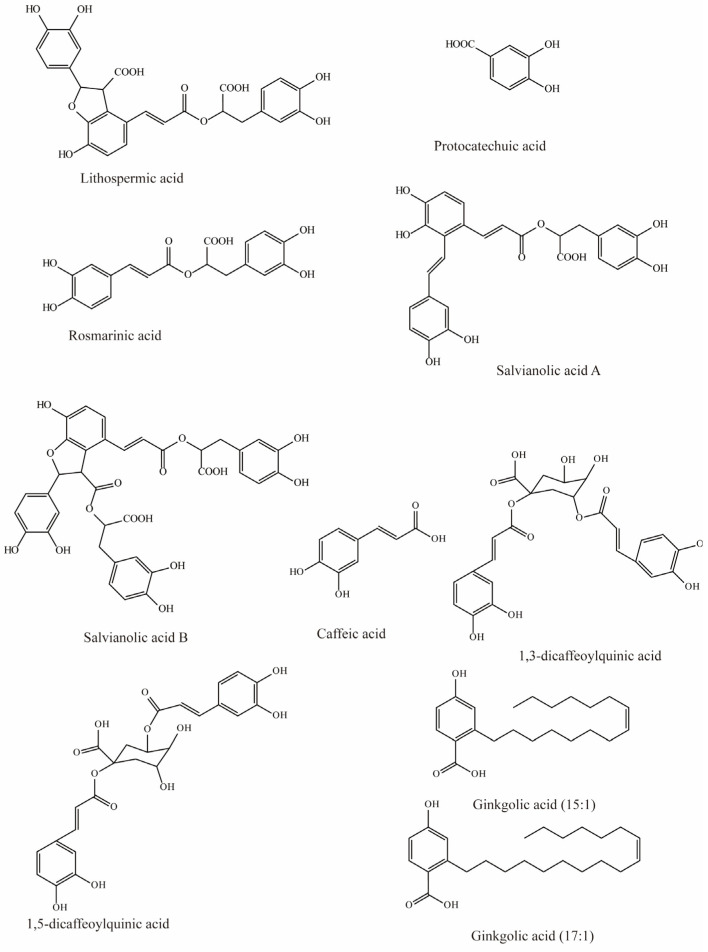
Structural diagram showing OAT3 inhibitors of phenolic acids.

**Figure 2 molecules-28-04740-f002:**
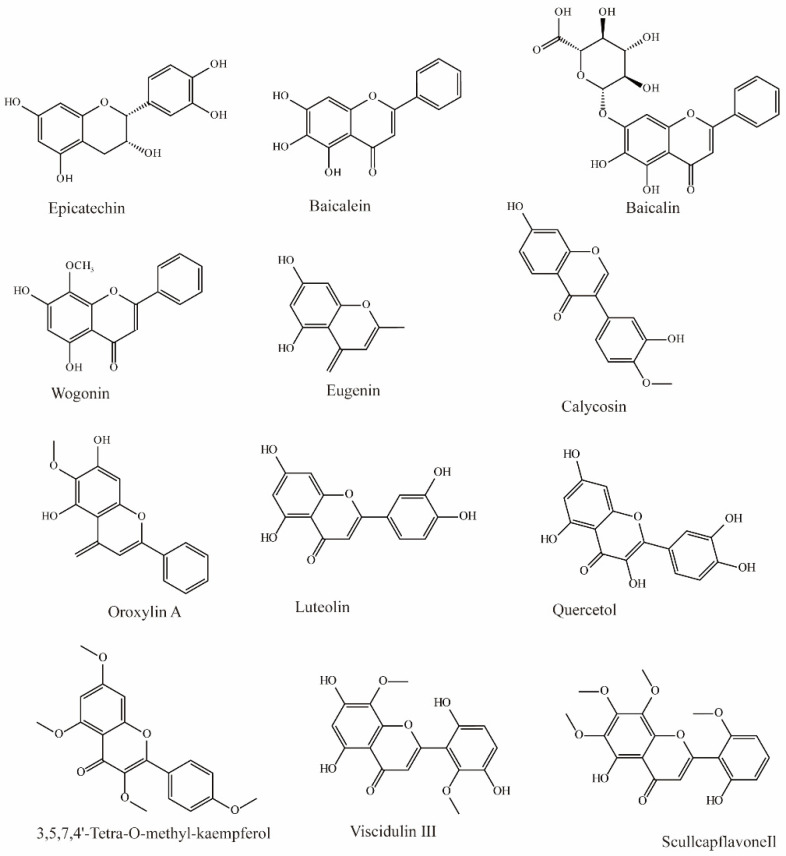
Structural diagram showing OAT3 inhibitors of flavonoids.

**Figure 3 molecules-28-04740-f003:**
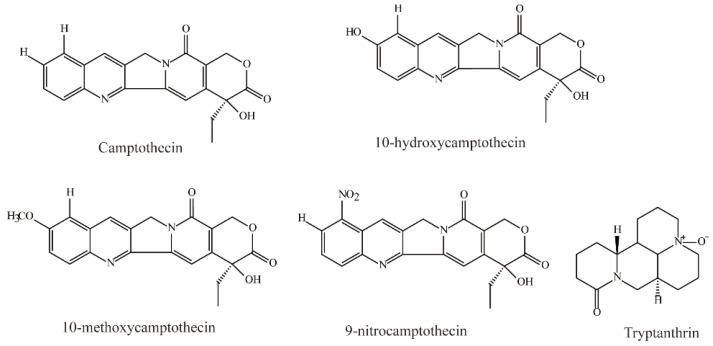
Structural diagram showing OAT3 inhibitors of alkaloids.

**Figure 4 molecules-28-04740-f004:**
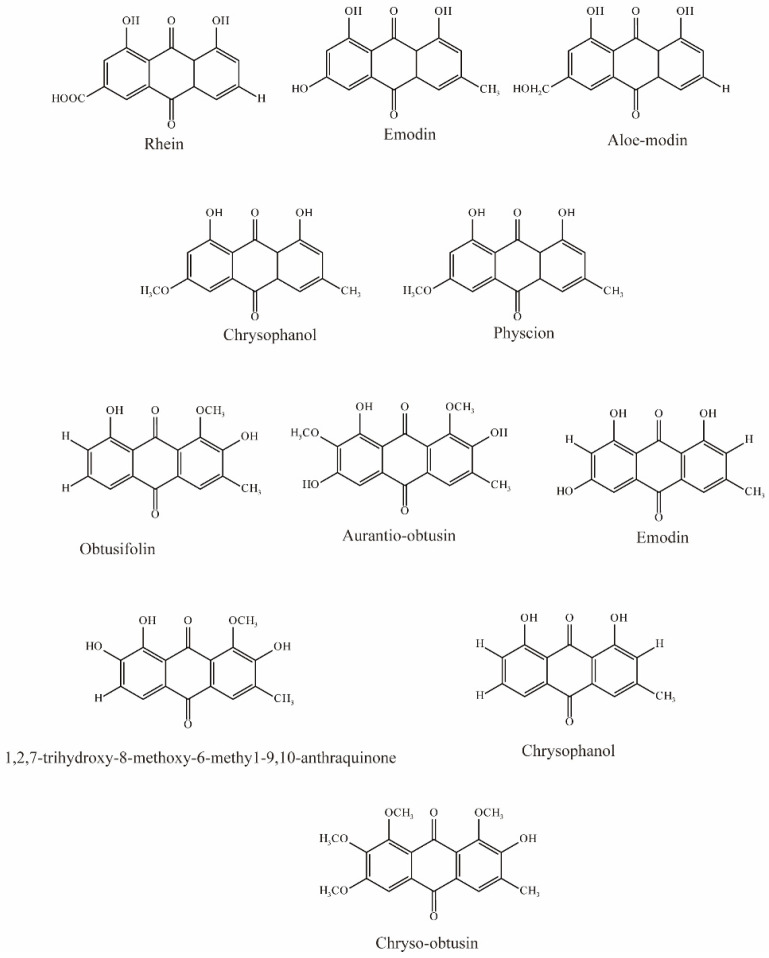
Structural diagram showing OAT3 inhibitors of anthraquinones.

**Figure 5 molecules-28-04740-f005:**
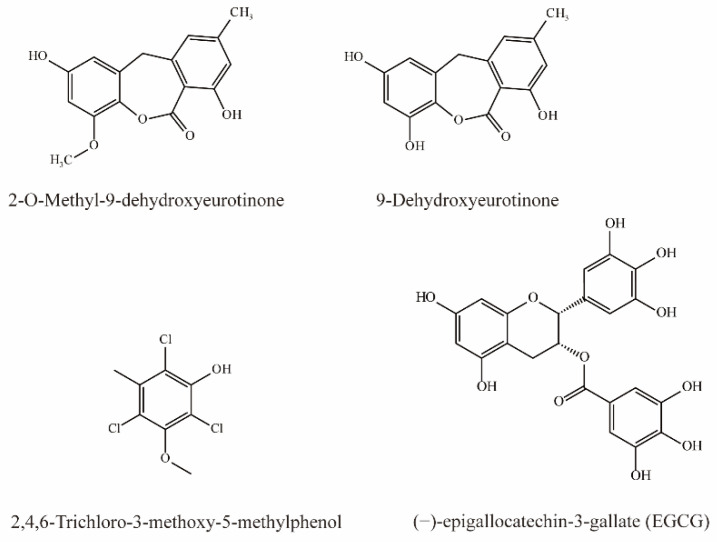
Structural diagram showing OAT3 inhibitors of phenols.

**Figure 6 molecules-28-04740-f006:**
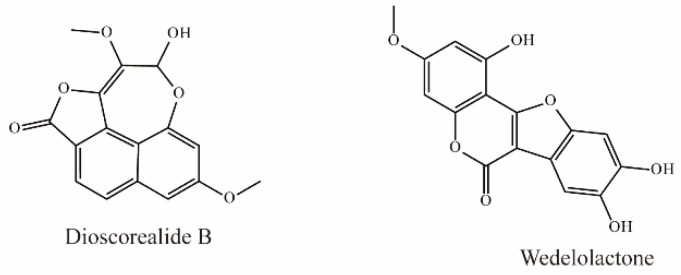
Structural diagram showing OAT3 inhibitors of phenylpropanoids.

**Figure 7 molecules-28-04740-f007:**
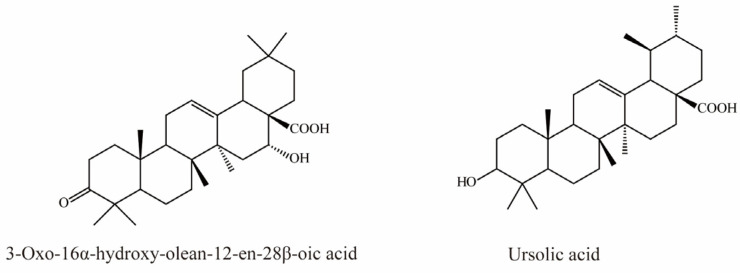
Structural diagram showing OAT3 inhibitor of terpenes.

**Figure 8 molecules-28-04740-f008:**
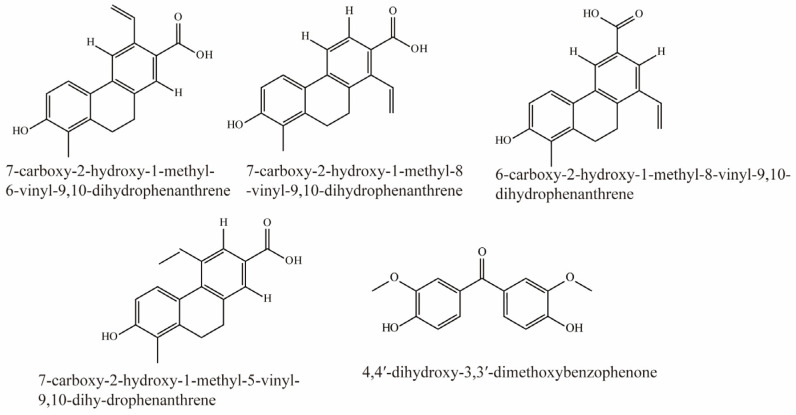
Structural diagram showing OAT3 inhibitors of phenanthrenoids.

**Table 1 molecules-28-04740-t001:** OAT3 and synthetic drug–drug interactions (DDIs).

Victim Drug	Perpetrator Drug	Model	IC_50_ (μM)	Effects	Reference
Imipenem	Cilastatin	Intravenous with cilastatin and imipenem 45 mg/kg in rats	-	AUC_0→∞_ ↑, T_1/2β_ ↑, CL_p_ ↓, CL_R_ ↓	[35]
Diclofenac	Cilastatin	hOAT3-HEK293 cells	411	-	[36]
		Diclofenac-induced acute kidney injury in mice	-	AUC_0–12h_ ↑, CL_p_ ↓, t_1/2_ ↑	[36]
Diclofenac acyl glucuronide	Cilastatin	hOAT3-HEK293 cells	813	-	[36]
		Diclofenac-induced acute kidney injury in mice	-	AUC_0–12h_ ↑, t_1/2_ ↑	[36]
MTX	Proton pump inhibitors	hOAT3-HEK293 cells	0.4–5.5	-	[37]
	Rhein	hOAT3-HEK293 cells	0.77	-	[38]
		Intravenous with MTX 5 mg/kg and/or rhein 1 mg/kg in rats	-	C_max_ ↑, AUC ↑, t_1/2β_ ↑, CL_P_ ↓	[38]
	Diclofenac-Glu	hOAT3-HEK293 cells	3.17	-	[39]
	R-Ibuprofen-Glu	hOAT3-HEK293 cells	60.1	-	[39]
	S-Ibuprofen-Glu	hOAT3-HEK293 cells	57.0	-	[39]
	R-Flurbiprofen-Glu	hOAT3-HEK293 cells	19.4	-	[39]
	S-Flurbiprofen-Glu	hOAT3-HEK293 cells	31.7	-	[39]
	R-Naproxen-Glu	hOAT3-HEK293 cells	129	-	[39]
	S-Naproxen-Glu	hOAT3-HEK293 cells	51.4	-	[39]
	Tranilast	Oral administration of MTX 5 mg/kg and tranilast 10 mg/kg in rats	-	C_max_ ↑, AUC_0–24h_ ↑, CL_z/f_ ↓, V_z/f_ ↓	[40]
Enalaprilat	Benzbromarone	hOAT3-HEK293 cells	0.14	-	[29]
	Glimepiride	hOAT3-HEK293 cells	0.37	-	[29]
	Febuxostat	hOAT3-HEK293 cells	0.41	-	[29]
	Verinurad	hOAT3-HEK293 cells	0.80	-	[29]
	Lesinurad	hOAT3-HEK293 cells	0.80	-	[29]
	Telmisartan	hOAT3-HEK293 cells	0.86	-	[29]
	Valsartan	hOAT3-HEK293 cells	0.90	-	[29]
	Sulfinpyrazone	hOAT3-HEK293 cells	0.95	-	[29]
	Repaglinide	hOAT3-HEK293 cells	2.10	-	[29]
	Gemfibrozil	hOAT3-HEK293 cells	3.50	-	[29]
Enalaprilat	Probenecid	hOAT3-HEK293 cells	5.12	-	[29]
	Diclofenac sodium	hOAT3-HEK293 cells	6.13		[29]
Quinaprilat	Gemcabene	hOAT3-HEK293 cells rOat3-HEK293 cell	35 48	-	[28]
		Oral administration of quinapril 3 mg/kg and gemcabene (30 mg/kg reduced to 10 mg/kg at day 3) in rats	-	AUC_0–24h_ ↑, CL_R_ ↓	[28]
	Gemcabene acylglucuronide	hOAT3-HEK293 cells rOat3-HEK293 cell	197 133	-	[28]
		Oral administration of quinapril 3 mg/kg and gemcabene (30 mg/kg reduced to 10 mg/kg at day 3) in rats	-	AUC_0–24h_ ↑, CL_R_ ↓	[28]
Bezafibrate	Mizoribine	Oral administration/intravenous with bezafibrate 20 mg/kg and/or mizoribine 15 mg/kg in rats	-	AUC ↑; t_1/2β_ ↑, CL_R_ ↓	[41]
Acyclovir	Benzylpenicillin	Intravenous with acyclovir 30 mg/kg and benzylpenicillin 30 mg/kg in rats	-	CL_R_ ↓, t_1/2β_ ↑	[42]
Tazobactam	Piperacillin	Intravenous with tazobactam 26.25 mg/kg and piperacillin 210 mg/kg in rats	-	CL_p_ ↓, CL_R_ ↓, AUC ↑, t_1/2β_ ↑, and k_m_ ↑	[43]

Total body clearance (CL/F); area under the concentration–time curve (AUC); maximum serum concentration (C_max_); plasma clearance (CL_p_); plasma/serum concentration half-time (t_1/2β_); renal clearance (CL_R_); ↓, reduced level; ↑, increased level; -, data were not reported.

**Table 2 molecules-28-04740-t002:** OAT3 and herb–drug interactions.

Drug	Herb	Indications for Herb	Model	IC_50_ (μM)	Effects	Reference
MTX	Puerarin	Hypertension, hyperlipidemia	hOAT3-HEK293 cells;	27.9	-	[57]
			Oral administration of MTX 5 mg/kg and PUR 50 mg/kg in rats	-	C_max_ ↑ AUC ↑, t_1/2β_ ↑; CL_p_ ↓	
Telmisartan	Steviol glucuronide	Hyperglycemia, hypertension, diabetes, inflammation	hOAT3-HEK293 cells	2.92	-	[58]
Diclofenac	Steviol glucuronide		hOAT3-HEK293 cells	8.01	-	[58]
Mulberrin	Steviol glucuronide		hOAT3-HEK293 cells	9.97	-	[58]
Probenecid	Steviol acyl glucuronide	Hypertension, diabetes, inflammation	hOAT3-HEK293 cells rOat3-HEK293 cells	4.9 2.3	-	[59]
			Oral administration of rebaudioside A 15 mg/kg and probenecid 20 mg/kg/glimepiride 5 mg/kg in rats	-	C_max_ ↑, AUC_6–8h_ ↑	[59]
Glimepiride	Steviol acyl glucuronide	Hyperglycemia, hypertension, diabetes, inflammation	hOAT3-HEK293 cells rOat3-HEK293 cells	0.81 0.77	- -	[59]
			Oral administration of rebaudioside A 15 mg/kg and probenecid 20 mg/kg/glimepiride 5 mg/kg in rats	-	C_max_ ↑; AUC_6–8h_ ↑	[59]
Imipenem	Apigenin	Viral infections, inflammation	hOAT3-HEK293 cells; Intravenous with imipenem 200 mg/kg and apigenin 10 mg/kg in rabbits	2.29	-	[60]
				-	AUC_0–6h_ ↑, CL_p_ ↓, CL_r_ ↓	[60]
Enalaprilat	Galangin	Hyperlipidemia, atherosclerosis, diarrhea	hOAT3-HEK293 cells	0.030	-	[29]
	Chrysin	Hyperlipidemia, diabetes, hypertension	hOAT3-HEK293 cells	0.044	-	[29]
	Kaempferol	Atherosclerosis, diabetes, inflammation	hOAT3-HEK293 cells	0.088	-	[29]
	Oroxylin A	Inflammation	hOAT3-HEK293 cells	0.22	-	[29]
	Wogonin	Inflammation	hOAT3-HEK293 cells	0.24	-	[29]
	Apigenin	Viral infections, inflammation	hOAT3-HEK293 cells	0.33	-	[29]
	Luteolin	Cardiovascular disease, hyperuricemia	hOAT3-HEK293 cells	0.66	-	[29]
	Gossypetin	Atherosclerosis	hOAT3-HEK293 cells	0.71	-	[29]
	Quercetin	Hypertension, hyperlipidemia	hOAT3-HEK293 cells	0.75	-	[29]
Enalaprilat	Mulberrin	Liver Fibrosis	hOAT3-HEK293 cells	1.22	-	[29]
	Eriodictyol	Inflammatory	hOAT3-HEK293 cells	1.48	-	[29]
	Fisetin	Liver cancer, inflammation	hOAT3-HEK293 cells	3.82	-	[29]
	Daidzein	Hypertension, ischemic cerebrovascular disease, hyperlipidemia	hOAT3-HEK293 cells	5.80	-	[29]
	Taxifolin	Cardiovascular disease, inflammatory	hOAT3-HEK293 cells	7.02	-	[29]
	Apigetrin	Oxidative stress	hOAT3-HEK293 cells	7.80	-	[29]
	Isoquercetin	Tumor, hypertension, hyperlipidemia, inflammation	hOAT3-HEK293 cells	11.71	-	[29]
	Wogonoside	Inflammatory	hOAT3-HEK293 cells	11.73	-	[29]
	Myricetin	Diabetes, cardiovascular disease	hOAT3-HEK293 cells	22.58	-	[29]
FS	*Juncus effusus*	Oedema, urinary discomfort	Oral administration of *Juncus effusus* extract 100 mg/kg and furosemide 10 mg/kg in rats	-	AUC_0–t_ ↑	[61]
	Rhubarb	Hepatitis, abdominal pain, constipation	Oral administration of rhubarb extract 5 g crude drug/kg/day and furosemide 10 mg/kg for 7 days in rats	-	AUC_0–t_ ↑	[62]

Total body clearance (CL/F); area under the concentration–time curve (AUC); maximum serum concentration (C_max_); plasma clearance (CL_p_); plasma/serum concentration half-time (t_1/2β_); renal clearance (CL_R_); ↓, reduced level; ↑, increased level; -, data were not reported.

**Table 3 molecules-28-04740-t003:** Natural active compounds with OAT3 inhibitory activity.

Category	Natural Active Compound	Source	Model	OAT3 Activity	Reference
Phenolic acids	Caffeic acid	-	rOat3-HEK293 cells (17.5 nM [^3^H]-Estrone sulfate as substrate)	IC_50_ = 5.4 μM	[71]
			Intravenous with caffeic acid at 5 mg/kg (25 mg/kg phenolsul fonphthalein as substrate) in rats	phenolsul fonphthalein CL_r_ ↓	
Flavonoids	Baicalein	*Scutellaria Baicalensis*	hOAT3-HEK293 cells (300 nM [^3^H]-Estrone sulfate as substrate)	IC_50_ = 2.4 μM	[72]
	Baicalin	*Scutellaria Baicalensis*	hOAT3-HEK293 cells (300 nM [^3^H]-Estrone sulfate as substrate)	IC_50_ = 13.0 μM	[72]
	Wogonin	*Scutellaria Baicalensis*	hOAT3-HEK293 cells (300 nM [^3^H]-Estrone sulfate as substrate)	IC_50_ = 1.3 μM	[72]
	Wogonin	Leaves of *Nelumbo nucifera Gaertn.*	hOAT3-HEK293 cells (10 μM 6-CF as substrate)	IC_50_ = 2.35 μM	[73]
			AAN model mice induced by AAI: intravenous with 5 mg/kg AAI and 100 mg/kg wogonin	AAI AUC_0–2h_ ↑, C_max_ ↑ CL/F ↓	
	Eugenin	Drynaria fortunei (Kunze) *J. Sm.*	hOAT3-HEK293 cells (10 μM 6-CF as substrate)	IC_50_ = 10.77 μM	[73]
	Calycosin	-	hOAT3-HEK293 cells (10 μM 6-CF as substrate)	IC_50_ = 11.44 μM	[73]
Flavonoids	Oroxylin A	*Nelumbo nucifera Gaertn.*	hOAT3-HEK293 cells (10 μM 6-CF as substrate)	IC_50_ = 1.68 μM	[73]
	Luteolin	*Eclipta prostrata* L.	hOAT3-HEK293 cells (10 μM 6-CF as substrate)	IC_50_ = 4.20 μM	[73]
	Quercetol	*Eclipta prostrata* L.	hOAT3-HEK293 cells (10 μM 6-CF as substrate)	IC_50_ = 7.27 μM	[73]
	3,5,7,4′-Tetra-O-methyl-kaempferol	*Epimedium brevicornu* Maxim.	hOAT3-HEK293 cells (10 μM 6-CF as substrate)	IC_50_ = 14.77 μM	[73]
	Viscidulin III	*Nelumbo nucifera Gaertn.*	hOAT3-HEK293 cells (10 μM 6-CF as substrate)	IC_50_ = 1.51 μM	[73]
	Scullcapflavone II	*Eclipta prostrata* L.	hOAT3-HEK293 cells (10 μM 6-CF as substrate)	IC_50_ = 6.70 μM	[73]
Alkaloids	Camptothecin	*Camptotheca acuminate* Decne.	hOAT3-HEK293 cells (300 nM [^3^H]-Estrone sulfate as substrate)	IC_50_ = 3.5 μM	[74]
	10-hydroxycamptothecin	*Camptotheca acuminate* Decne.	hOAT3-HEK293 cells (300 nM [^3^H]-Estrone sulfate as substrate)	IC_50_ = 4.6 μM	[74]
	10-methoxycamptothecin	*Camptotheca acuminate* Decne.	hOAT3-HEK293 cells (300 nM [^3^H]-Estrone sulfate as substrate)	IC_50_ = 4.8 μM	[74]
	9-nitrocamptothecin	*Camptotheca acuminate* Decne.	hOAT3-HEK293 cells (300 nM [^3^H]-Estrone sulfate as substrate)	IC_50_ = 2.8 μM	[74]
	Tryptanthrin	-	hOAT3-HEK293 cells (300 nM [^3^H]-Estrone sulfate as substrate)	IC_50_ = 0.93 μM	[70]
Anthraquinone	Rhein	Rhubarb	hOAT3-HEK293 cells (ranging from 0.5 μM to 200 μM Fluo as substrate)	IC_50_ = 0.08 μM	[62]
	Emodin	Rhubarb	hOAT3-HEK293 cells (ranging from 0.5 μM to 200 μM Fluo as substrate)	IC_50_ = 1.22 μM	[62]
	Aloe-modin	Rhubarb	hOAT3-HEK293 cells (ranging from 0.5 μM to 200 μM Fluo as substrate)	IC_50_ = 5.37 μM	[62]
	Obtusifolin	*Semen cassiae* *Semen cassiae*	hOAT3-HEK293 cells (5 μM 6-CF as substrate)	IC_50_ = 1.71 μM	[75]
	Aurantio-obtusin		hOAT3-HEK293 cells (5 μM 6-CF as substrate)	IC_50_ = 1.23 μM	[75]
Anthraquinone	Emodin	*Semen cassiae*	hOAT3-HEK293 cells (5 μM 6-CF as substrate)	IC_50_ = 2.06 μM	[75]
	1,2,7-trihydroxy-8-methoxy-6-methy1-9,10-anthraquinone	*Semen cassiae*	hOAT3-HEK293 cells (5 μM 6-CF as substrate)	IC_50_ = 3.44 μM	[75]
	Chrysophanol	*Semen cassiae*	hOAT3-HEK293 cells (5 μM 6-CF as substrate)	IC_50_ = 6.20 μM	[75]
	Chryso-obtusin	*Semen cassiae*	hOAT3-HEK293 cells (5 μM 6-CF as substrate)	IC_50_ = 7.46 μM	[75]
Phenols	2-*O*-methyl-9-dehydroxyeurotinone	*Semen cassiae* *Semen cassiae*	hOAT3-HEK293 cells (5 μM 6-CF as substrate)	IC_50_ = 7.14 μM	[75]
	9-dehydroxyeurotinone		hOAT3-HEK293 cells (5 μM 6-CF as substrate)	IC_50_ = 7.35 μM	[75]
	(−)-epigallocatechin-3-gallate(EGCG)	Green tea	hOAT3-HEK293 cells (5 μM 6-CF as substrate), rOat3 -HEK293 cells (5 μM 6-CF as substrate)	K_i_ = 162 μM K_i_ = 56.5 μM	[76]
			Intravenous with 6-CF 1 mg/kg and EGCG 60 mg/kg in rats	6-CF AUC_0−60min_ ↑, CLr ↓, BNU ↑	
	2,4,6-Trichloro-3-methoxy-5-methylphenol	-	hOAT3-HEK293 cells (10 μM 6-CF as substrate)	IC_50_ = 3.93 μM	[73]
Phenylpropanoids	Dioscorealide B	*Dioscorea* *septemloba Thunb.*	hOAT3-HEK293 cells (10 μM 6-CF as substrate)	IC_50_ = 7.73 μM	[73]
	Wedelolactone	*Eclipta prostrata* L.	hOAT3-HEK293 cells (10 μM 6-CF as substrate)	IC_50_ = 7.09 μM	
Terpenes	3-Oxo-16α-hydroxy-olean-12-en-28β-oic acid	-	hOAT3-HEK293 cells (10 μM 6-CF as substrate)	IC_50_ = 8.05 μM	[73]
	Ursolic acid	Pomegranate	hOAT3-HEK293 cells (300 nM [^3^H]-Estrone-3-sulfate as substrate)	IC_50_ = 18.9 μM	[77]
Others	*Anchusa azurea* (A)	-	hOAT3-HEK293 cells (10 μM 6-CF as substrate)	IC_50_ = 3.836 µg/mL	[61]
	*Anchusa azurea* (B)	-	hOAT3-HEK293 cells (10 μM 6-CF as substrate)	IC_50_ = 0.343 µg/mL	[61]
	*Symphytum asperum* (B)	-	hOAT3-HEK293 cells (10 μM 6-CF as substrate)	IC_50_ = 0.406 µg/mL	[61]
Others	*Echium russicum*(B)	-	hOAT3-HEK293 cells (10 μM 6-CF as substrate)	IC_50_ = 0.460 µg/mL	[61]
	*Juncus effusus* (D)	-	hOAT3-HEK293 cells (10 μM 6-CF as substrate)	IC_50_ = 1.210 µg/mL	[61]
	*Polygonum hydropiper* (D)	-	hOAT3-HEK293 cells (10 μM 6-CF as substrate)	IC_50_ = 1.296 µg/mL	[61]
	*Geranium tuberosum* (B)	-	hOAT3-HEK293 cells (10 μM 6-CF as substrate)	IC_50_ = 1.878 µg/mL	[61]
	*Primula macrocalyx* (D)	-	hOAT3-HEK293 cells (10 μM 6-CF as substrate)	IC_50_ = 2.081 µg/mL	[61]
	*Glycyrrhiza glabra* (D)	-	hOAT3-HEK293 cells (10 μM 6-CF as substrate)	IC_50_ = 2.399 µg/mL	[61]
	*Polygonum hydropiper* (D)	-	hOAT3-HEK293 cells (10 μM 6-CF as substrate)	IC_50_ = 2.966 µg/mL	[61]
	*Juniperus oblonga* (*LF + FR*) (D)	-	hOAT3-HEK293 cells (10 μM 6-CF as substrate)	IC_50_ = 3.047 µg/mL	[61]
	*Juniperus oblonga* (*LF + FR*) (B)	-	hOAT3-HEK293 cells (10 μM 6-CF as substrate)	IC_50_ = 1.117 µg/mL	[61]
	*Astracantha microcephala* (D)	-	hOAT3-HEK293 cells (10 μM 6-CF as substrate)	IC_50_ = 4.619 µg/mL	[61]
	*Chaerophyllum bulbosum* (H)	-	hOAT3-HEK293 cells (10 μM 6-CF as substrate)	IC_50_ = 4.791 µg/mL	[61]
Phenanthrenoids	7-carboxy-2- hydroxy-1-methyl-6-vinyl-9,10-dihydrophenanthrene	*Juncus effusus.*	hOAT3-HEK293 cells (4 μM 6-CF as substrate)	IC_50_ = 1.3 μM	[78]
	7-carboxy-2- hydroxy-1-methyl-8-vinyl-9,10-dihydrophenanthrene	*Juncus effusus.*	hOAT3-HEK293 cells (4 μM 6-CF as substrate)	IC_50_ = 1.1 μM	[78]
	6-carboxy-2-hydroxy-1-methyl-8- vinyl-9,10-dihydrophenanthrene	*Juncus effusus.*	hOAT3-HEK293 cells (4 μM 6-CF as substrate)	IC_50_ = 2.8 μM	[78]
	7-carboxy-2-hydroxy-1-methyl-5-vinyl-9,10-dihy-drophenanthrene	*Juncus effusus.*	hOAT3-HEK293 cells (4 μM 6-CF as substrate)	IC_50_ = 1.1 μM	[78]
	4,4′-dihydroxy-3,3′-dimethoxybenzophenone	*Juncus effusus.*	hOAT3-HEK293 cells (4 μM 6-CF as substrate)	IC_50_ = 2.3 μM	[78]

Total body clearance (CL/F), area under the concentration–time curve (AUC), maximum serum concentration (C_max_), plasma clearance (CL_p_), plasma/serum concentration half-time (t_1/2β_), renal clearance (CL_R_), ↓ reduced level, ↑ increased level. H represents hexane extract, D for dichloromethane extract, B for n-Butanol extract, and A for aqueous phase extract.

## Data Availability

All relevant data are included in this article.

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
