# Peer review of "Recent Advances in Synthetic Drugs and Natural Actives Interacting with OAT3"

_molecules, 2023, doi:10.3390/molecules28124740_

Round 1

Reviewer 1 Report

The Review form Chen et al. entitled “Recent advances in synthetic drugs and natural actives interacting with OAT3” contribute a great summery for understanding the role of OAT3 in pharmacokinetics particularly in the kidney and involvement in DDI as well as DHI. The physiological, pathophysiological as well as Pharmacological impact of OAT3 are well described. This manuscript emphasize or summarize the essential contribution of OAT3 in beneficial and risks or disadvantage by DDI or DHI. There review summarize data generated over decades from different scientific experiments in vivo as well as in vitro in very clear structured tables and statements. Therefore, it is a great contribution for the scientific community dealing with the interaction of drugs and natural products with specific and clinical relevant transporter as OAT3.

I suggest the editorial board to accept the review for publication.

The Review form Chen et al. entitled “Recent advances in synthetic drugs and natural actives interacting with OAT3” contribute a great summery for understanding the role of OAT3 in pharmacokinetics particularly in the kidney and involvement in DDI as well as DHI. The physiological, pathophysiological as well as Pharmacological impact of OAT3 are well described. This manuscript emphasize or summarize the essential contribution of OAT3 in beneficial and risks or disadvantage by DDI or DHI. There review summarize data generated over decades from different scientific experiments in vivo as well as in vitro in very clear structured tables and statements. Therefore, it is a great contribution for the scientific community dealing with the interaction of drugs and natural products with specific and clinical relevant transporter as OAT3.

I suggest the editorial board to accept the review for publication.

Author Response

Dear Reviewer,

We are very honored to get your approval.

Thanks for spending time and effort to review this manuscript.

Thank you, and best regards. 

Yours sincerely, 

Ying Chen 

Reviewer 2 Report

Chen and colleagues prepared a review on synthetic drugs and natural actives that interacted with organic anion transporter 3 (OAT3). As stated in their abstract and introduction, their focus is drug-drug interactions and herbal-drug interactions mediated by OAT3 and inhibitors of OAT3 in natural active compounds. The review covers from 1998–2023.

I recommended that this manuscript can be accepted for publication with minor revision. Further efforts are required for improving the quality, grammar and coherence of the manuscript.

I have a few comments about the visual perception of this review. The authors rather carelessly approached its design and made mistakes that reduce its quality and require immediate elimination. So, in the text of the manuscript there are not enough spaces (line 25, 46, 55, 57, 58, etc.). Misprints were made (line 24, 319, etc.), extra punctuation marks were added (line 107, 320, etc.). Further efforts are required for improving the quality, grammar and coherence of the manuscript. In Table 3 and Figure 1, ursolic acid is classified as a phenolic acid, which is incorrect and should be moved to the terpene class since ursolic acid is a pentacyclic triterpenoid. In figure 7, the structural formula of 3-Oxo-16α-hydroxy-olean-12-en-28β-oic acid is incorrect (see ursolic acid formula). Table 1 also lists the class others and lists compounds 1,2,3,10,16 without their names. I think it is necessary to name them according to the IUPAC nomenclature and call the class of compounds phenanthrenoids, as they are called in the literature reference 78 to which the authors refer. By the way, this reference also has names for these compounds. In accordance with this, the title of Fig. 8 should be corrected. In the conclusion section, future perspectives should be added to highlight the importance of the study and the future research that needs to be done.

Reviewer 3 Report

In this review, Chen and colleagues present a fairly comprehensive review of drug-drug, herbal-drug and natural product inhibitors of OAT3.  The review is well-written and documented with just a few minor areas of clarification needed, particularly surrounding the Tables.

1) In Table 1, more description is needed for what is measured in cell-based assay and recorded as effect.  What does "-" mean?  This all should be included in the Table Legend.

2) Page 5, second paragraph - Is the reduced cytotoxicity observed with diclofenac acyl glucuronide also seen with diclofenac itself?  Please include some discussion of this.

3) Page 8, line 147 - where was the survey conducted?  There may be country-specific differences in natural products usage.  A discussion of differences may be warranted if they exist.

4) Table 2 - same comment as for Table 1.  Also, a column where herb most commonly found/used and for any particular indication would be useful and informative.

5) Table 3 - Is "effect" the IC50 for inhibition of OAT3 by natural substance?  Please clarify in Table. 

6) With regards to effects of natural bioactive compounds, a discussion of whether exposures (PK) typically observed are sufficient to cause effects based on the reported IC50 might be helpful.  Although it is understood this would be caveated by potential differences in protein binding and tissue specific concentrations.  

7) Conclusions - the authors conclude with the generalized statement that co-administration of OAT3 inhibition may increase drug accumulation in kidney and cause nephrotoxicity, yet nearly all examples presented indicated a reduction in nephrotoxicity.  Can the authors comment on this?  Is a modification of the conclusions warranted?

8) Minor editorial note - the pagination is off such that all pages subsequent to 16 all say "16 of 23"

Occasionally, there is incorrect use of an adverb (with ...ly) when it should be an adjective.  Example page 5 line 121 "concomitantly" should be "concomitant"
